# Enhancing the Production of Hydrolytic Enzymes in Elicited Tamarillo (*Solanum betaceum* Cav.) Cell Suspension Cultures

**DOI:** 10.3390/plants12010190

**Published:** 2023-01-02

**Authors:** Bruno Casimiro, Inês Mota, Paula Veríssimo, Jorge Canhoto, Sandra Correia

**Affiliations:** 1Centre for Functional Ecology, TERRA Associate Laboratory, Department of Life Sciences, Calçada Martim de Freitas, University of Coimbra, 3000-456 Coimbra, Portugal; 2Center for Neuroscience and Cell Biology, University of Coimbra, Calçada Martim de Freitas, 3000-456 Coimbra, Portugal; 3InnovPlantProtect CoLab, Estrada de Gil Vaz, 7351-901 Elvas, Portugal

**Keywords:** cell growth, chitosan, elicitation, glycosidase activity, hydrolytic enzymes

## Abstract

Plant cell suspension cultures are widely used as a tool for analyzing cellular and molecular processes, metabolite synthesis, and differentiation, bypassing the structural complexity of plants. Within the range of approaches used to increase the production of metabolites by plant cells, one of the most recurrent is applying elicitors capable of stimulating metabolic pathways related to defense mechanisms. Previous proteomics analysis of tamarillo cell lines and cell suspension cultures have been used to further characterize and optimize the growth and stress-related metabolite production under in vitro controlled conditions. The main objective of this work was to develop a novel plant-based bioreactor system to produce hydrolytic enzymes using an elicitation approach. Based on effective protocols for tamarillo micropropagation and plant cell suspension culture establishment from induced callus lines, cell growth has been optimized, and enzymatic activity profiles under in vitro controlled conditions characterized. By testing different sucrose concentrations and the effects of two types of biotic elicitors, it was found that 3% (*w/v*) sucrose concentration in the liquid medium enhanced the production of hydrolytic enzymes. Moreover, casein hydrolysate at 0.5 and 1.5 g/L promoted protein production, whereas yeast extract (0.5 g/L) enhanced glycosidase activity. Meanwhile, chitosan (0.05 and 0.1 g/L) enhanced glycosidases, alkaline phosphates, and protease activities.

## 1. Introduction

Over the last decades, plants have been efficiently used as platforms for the rapid and scalable production of functional recombinant proteins [1]. Plant-based production of recombinant proteins (e.g., vaccines, therapeutics, diagnostic compounds, industrial enzymes) is now a rapidly developing segment of the bioeconomy, with recent data showing how it can contribute to global responses in emergency scenarios [2], as was the case for the rapid production of SARS-CoV-2 proteins for COVID-19 research by the transient transformation of tobacco plants [3].

Within the available options for Molecular Farming platforms, plant cell suspensions (PCSs) are becoming prominent as sustainable and efficient systems for producing high-quality molecules following Good Manufacturing Practice standards [1]. PCSs have several key advantages over conventional production systems, including rapid growth, improved consistency of the protein product with the use of controlled bioreactors which are less prone to biotic- and abiotic-induced variations, and the fact that plant cells do not harbor any known human pathogens [4,5]. Unlike microbial cells, they can perform complex post-translational modifications, such as glycosylation, needed for the correct folding and stability of many proteins [6]. Due to simple growth conditions, PCSs cultures offer a low-cost alternative to mammalian production systems. In addition, their controlled and confined growth using classical fermentation technology overcome regulatory and environmental concerns regarding the potential release of genetically modified organisms (GMOs) linked to whole-plant systems [7]. One of the most important advantages of using PCSs over whole plants is that the procedures for the isolation and purification of the products are much more efficient since tagged recombinant proteins can be secreted into the culture medium [8].

Tamarillo (*Solanum betaceum* Cav.) is a solanaceous tree native to the Andean region producing edible fruits widely used in South American countries, with an emerging economic relevance due to their nutritional properties [9]. This species has become a reliable system to investigate plant development and biotechnological processes [10] for which optimized in vitro culture protocols are available [10]. Starting from mature zygotic embryos or young leaves tamarillo-induced callus lines (ICL) can be obtained under the presence of exogenous auxins and high sucrose concentrations [10]. 

Analysis of ICL cellular protein profiles revealed high expression levels of stress-related proteins [11,12], confirming the role of tissue culture systems in inducing molecular reprogramming. During recent experiments, *S. betaceum* ICL was grown efficiently as PCSs [13]. The evaluation of their extracellular protein patterns showed that cell suspensions of tamarillo produce and secrete a wide array of proteins into the culture medium, including endoglucanases, proteases, and low molecular weight (>20 kDa) peptides [13]. These results followed the previously obtained intracellular protein profiles of tamarillo ICL [11]. 

β-Glucan is an eco-friendly, biodegradable, and economical biopolymer very effective against pathogens by eliciting plant innate immunity [14]. β-Glucan triggers the downward signaling cascade/s, resulting in the accumulation of different pathogenesis-related proteins, reactive oxygen species (ROS), antioxidant defense enzymes, Ca^2+^-influx as well as activation of the mitogen-activated protein kinase (MAPK) pathway [14,15]. The most popular industrial uses are as a thickening and gelling agent in the food industry and as an ingredient in cosmetic and personal care products due to their soothing, moisturizing and anti-irritant properties [16]. 

Chitinases are antifungal proteins that interact with chitin biopolymer leading to the degradation of chitin by hydrolysis of the glycosidic bonds between two or more carbohydrates. They play a vital role in biological control and plant defense mechanisms against fungi [17]. Chitinases can be exploited as food preservatives, thereby increasing the shelf life of fruits and vegetables. Additionally, they can potentially be used in human medicines [18]. An important example of medical use for chitinases is the ability to increase the activity of antifungal and anti-tumor drugs such as chitohexaose and chitoheptaose [18]. Several reports have described the synergic interaction effect of chitinases and β-1,3-glucanases in antifungal defense in vitro and in vivo [19]. Proteases, namely aspartic, serine, and metalloproteases, have a wide range of applications in the food and beverage industry, such as the cheese, bakery, beer and wine industries [20].

For their importance, the increase in the production yields of these hydrolytic enzymes is a challenge that can be tackled with the usage of biotechnological approaches.

Elicitation is a commonly used strategy to enhance cell proliferation and stress-induced mechanisms that ultimately lead to optimized PCSs growth and the production of bioactive compounds [21,22]. 

Common elicitors are casein hydrolysate (CH), a complex mixture of 18 amino acids, vitamins, calcium, phosphate, and several microelements, and yeast extract (YE), a vitamin β-complex-rich elicitor, containing chitin, N-acetyl-glucosamine oligomers, β-glucan, ergosterol, and glycopeptides [23]. These organic substances are the active source of several amino acids, hormones, vitamins, fatty acids, carbohydrates, and other plant growth substances essential for plants’ proper growth and regeneration [24]. Chitosan (CTS), a deacetylated chitin, has often been used as an elicitor. It is derived from the cell wall of fungi and mimics the effects of several pathogenic fungi in stimulating defenses related to secondary metabolite production in plants [25,26,27]. This compound was found to induce plant defense responses by triggering cell growth and secondary metabolism and is frequently used as a biotic elicitor to improve secondary metabolite production [28,29,30]. 

The main objective of this work was to develop a novel plant-based bioreactor system to produce hydrolytic enzymes using an elicitation approach. To achieve this goal, the elicitation effects of CH, YE, and CTS were evaluated on the growth of two tamarillo ICL, and the intracellular protein production and enzymatic activity were evaluated under controlled and elicited conditions. The results obtained will set the basis for establishing and optimizing tamarillo cell suspension cultures as valuable tools for analyzing stress-related metabolic networks and as a promising system for plant-derived metabolite production in scaled-up approaches, such as bioreactors. 

## 2. Results

### 2.1. Cell Suspension Cultures Growth and Enzymatic Activity under Different Sucrose Concentrations

The two ICL tested (L1 and L2) were previously established in vitro in a solid MS medium with a sucrose concentration of 9% (*w/v*). When the cell suspension cultures were established, the effect of sucrose concentration was tested. So, 500 mg (FW) of each callus were placed inside flasks containing MS medium: three replicas with 9% (*w/v*) sucrose and another three with 3% (*w/v*) sucrose (Figure 1a). During cellular growth, some cytological changes were observed in the cells, similar for both lines. In the early stages, on day 3 of the culture, the cells were mostly isodiametric (Figure 1b) whereas beyond day 11 (Figure 1c,d), they were more elongated. Moreover, the cells often appeared plasmolyzed, in particular at the later stages of culture.

The optical density was used to evaluate the cell growth, which allowed the construction of growth curves for these two cell lines with sucrose concentration applied (Figure 2a).

L1 and L2 growth curves (Figure 2a) are quite similar. During the cellular growth, performed for 14 days, both lines demonstrated higher optical density values with 9% (*w/v*) of sucrose. In both cases, curves showed an initial phase with the lowest optical values (first day until day 7) and a second phase with an increase in those values (from day 7 to 14). According to the optical density values, since day 7, there has been a higher increase in the culture’s cellular growth. 

As can be seen in Figure 2b, the protein quantity by mg of callus was higher for the two cell lines with 9% (*w/v*) sucrose, but the specific production of proteases (Figure 2c left), glycoside hydrolases, and alkaline phosphatases (Figure 2c right) was higher with 3%(*w/v*) sucrose, in particular for L2. 

As the main objective of this preliminary assay was to understand the cellular growth dynamics and determine the best conditions which enhance the hydrolases’ specific production, the subsequent assays, with elicitors supplementation, were performed with the 3% (*w/v*) sucrose concentration.

### 2.2. Effect of Elicitors on the Growth of Cells Suspension Cultures

The two cell suspension cultures (L1 and L2) were grown with three different biotic elicitors: casein hydrolyzed (CH), yeast extract (YE), and chitosan (CTS). 

Growth curves relative to control for both cell lines (Figure 3) show that the cell growth increased with CH, particularly in L1 (Figure 3a,c), in which the number of cells was two times higher than the control. 

### 2.3. Elicited Cell Suspension Total Protein Production

The protein quantity production of L1 and L2 cell suspension cultures relative to control was evaluated (Figure 4). 

In L1, CH at 0.5 g/L promoted a statistically significant increase in three-fold of the protein quantity relative to the control, whereas 1.5 g/L promoted the growth of 1.7-fold. YE did not influence the protein quantity relative to the control, and CTS had a negative influence, especially with 0.05 g/L, with a statistically significant decrease of 0.9-fold relative to the control.

In L2, CH also promoted the increase in the protein quantity relative to the control, but in this case, only with the 1.5 g/L concentration (a statistically significant increase of two-fold relative to the control). YE and CTS negatively affect the protein quantity relative to control, especially CTS, with a statistically significant decrease of 0.7 and 0.9-fold for 0.05 g/L and 0.1 g/L, respectively.

### 2.4. Specific Enzymatic Activity

The specific enzyme production capacities of L1 and L2 post-elicitation were evaluated. Their enzymatic profiles were assayed using specific activity assays. The first group of specific substrates (Figure 5a–c) was used to evaluate the presence of proteases. The second group of substrates (Figure 5d–f) was used to evaluate the presence of glycoside hydrolases (MU-G, MU-NAG, and MU-C) and alkaline phosphatases (MU-P). The choice of these substrates is based on previous works with tamarillo cell suspension cultures [13].

The results are presented as the clustering of the specific activity obtained for each elicitor treatment for each cell line (Figure 5). The protease-specific activity was detected with all tested fluorescent substrates, but a statistically significant increase in the specific activity was detected in CTS-elicited cultures, particularly with 0.05 g/L in L1 and 0.1 g/L in L2. 

The data showed that the specific activity was statistically significantly higher, with YE at 1.5 g/L and CTS at 0.1 g/L in L2 for glycoside hydrolases. Alkaline phosphatase specific activity was statistically significantly higher with CTS at both concentrations in L1 and CTS 0.1 g/L in L2.

## 3. Discussion

The need for more sustainable and eco-friendly agricultural practices has been a driving force for plant biotechnology, especially in plant bioactive compounds research, by efficiently establishing plant stress under controlled conditions. To that aim, plant tissue culture, particularly PCSs, stands out as a reliable methodology, enabling the efficient production of metabolites that can act on protection against pathogenic and climatic threads [31].

In the first step, the cellular growth was evaluated for 14 days. When we look at the cell morphology variation during the growth, it can be seen that the cells become elongated in the later stages of the suspension cultures, which can be explained by the nutrient limitation associated with a low cell division rate [32] and a response to the hydrodynamic stress resulting from the shear force [33]. 

Sucrose, a carbon source, is an essential substrate for producing energy used in primary and secondary metabolism [34]. Though, an increase in the levels of sucrose has two different effects on cell suspension cultures: the first effect is the increase in osmotic pressure, altering the cellular environment, and the second one is the intensification of accessible carbohydrate sources as substrates in the liquid medium [35]. We have found that a liquid medium with 9% (*w/v*) sucrose is more effective at promoting cell growth than 3% (*w/v*) sucrose for both cell lines. Several studies have been made to relate cell growth with the initial sucrose concentration supply. However, in these studies, an elevated sucrose concentration demonstrated a decrease in cell growth. In cultured *Daucus carota* cells (carrot), sucrose concentrations above 7.5% (*w/v*) reduced cell growth [36]. Moreover, in cultures of *Gymnema sylvestre* with concentrations of 1% (*w/v*) and 3% (*w/v*) sucrose, fresh and dry weight (cellular density) were higher than for cultures with 5.7 or 9% (*w/v*) sucrose [37]. In these cases, higher sucrose concentrations resulted in an osmotic pressure affecting cell growth. In *Hancornia speciosa,* it was observed that the biomass and accumulation of desirable secondary metabolites were favored by a 3% (*w/v*) sucrose concentration [38]. In the tested tamarillo ICL, there was not a decline in cell growth with 3% sucrose (*w/v*), just a slow growth compared to the cultures grown with 9% sucrose.

The determination of the timing of the exponential phase was crucial since there is evidence that the elicitation strategy is more effective at the beginning of the stationary growth phase [26,39].

The cell growth dynamics for both lines are similar. During the 14 days, we observed an initial growth phase with the lowest optical values (until day 7) and a second phase with an increase in those values (from day 7 until day 14). Based on these data, it was predicted that the exponential phase, occurred between the seventh and 14th day.

Despite the limitations of PCSs growth evaluation based on optical density measurements [40], it can be useful as a preliminary step in the understanding of cell suspension dynamics. The subsequent growth dynamics observed in the elicitation experiment confirmed the described growth dynamics by cell counting. 

The amount of protein by mg of callus was higher for the two cell lines with 9% (*w/v*) sucrose. However, when the specific enzymatic activity related to the plant cell defense mechanisms was the factor analyzed, the detection for the proteases, glycoside hydrolases, and alkaline phosphatases was higher with 3% (*w/v*) sucrose, in particular for L2. The main goal of this work was to enhance the specific enzymatic production related to the cell defense mechanisms, and for that, the 3% (*w/v*) sucrose concentration was selected for the following experiments.

Elicitation is one of the most effective techniques for improving the biotechnological production of secondary metabolites [26]. Elicitors are inductive stress agents, of biotic or abiotic origins, with the ability to cause significant changes in the expression of plant defense-related genes, which may lead to the accumulation of secondary metabolites [41]. In our assays, the biotic elicitors used were divided into two groups, considering their effect in the cell cultures directed: CH and YE directed for cell growth and protein production [42,43,44,45] and CTS directed the promotion of plant defense mechanisms and the consequent increase in enzymatic activity [28,46,47]. 

Elicitor concentration is essential for elicitation, and the optimum level depends on the plant species [48]. According to Park et al. [49], high concentrations of elicitors trigger the hypersensitive response and lead to cell death. Thus, for CH and YE, two concentrations were tested 0.5 g/L and 1.5 g/L. For chitosan, the two tested concentrations were 0.05 g/L and 0.1 g/L. 

It was expected from previous works with *Oryza sativa* L. [42], *Corylus avellana* L. [43], *Stevia rebaudiana* [44], and *Panax vietnamensis* [50] that the cell growth at the concentrations used in our work would be promoted by CH and YE, which we found for CH when compared to control, especially for L1. Additionally, according to previous works with *Scrophularia striata* Boiss [51], CTS was expected to negatively affect cell growth and viability. In this study, we found that CTS had a neutral or negative impact on both lines’ growth compared to the control.

Concerning the elicited cell suspension protein production, the results presented align with the expected outcomes from previous research [51,52,53,54] and justify our choice of elicitors described above. CH promotes global protein production in cell suspension cultures and CTS, and a more specific enzymatic response related to the plant defense mechanisms. The protease-specific activity was detected with a statistically significant increase in CTS-elicited cultures, particularly with 0.05 g/L in L1 and 0.1 g/L in L2. 

The data also showed a specific activity with a statistically significantly higher detection with YE at 1.5 g/L in L1 and CTS 0.1 g/L in L2 for glycoside hydrolases, hence for alkaline phosphatase-specific activity, a statistically significantly higher detection was observed with CTS at both concentrations in L1 and CTS 0.1 g/L in L2.

As reviewed by Jakubas et al. [54], Dalvi et al. [55] and as demonstrated in *Hordeum vulgare* suspension cultures [56], CTS has a big impact on the plant defense mechanisms and secondary metabolite production and in particular the production of chitinases and β-1,3 glucanases. With these data, we can confirm that despite lower protein production, the CTS elicited tamarillo cell suspension cultures, and as a result, with the tested concentrations, tend to show a higher specific enzymatic production related to their defense mechanisms, particularly proteases, glycoside hydrolases, and alkaline phosphatase production.

## 4. Materials and Methods

### 4.1. Plant Material and Culture Conditions

#### 4.1.1. Callus Induction and Maintenance

The ICLs used in this study were obtained according to [10]. Briefly, young leaves from *Solanum betaceum* were placed in an induction medium composed of MS [57] basal medium (Duchefa Biochimie © Haarlem, The Netherlands) supplemented with 9% (*w/v*) sucrose, picloram (5 mg/L) and 2.5 g/L of Pytagel TM (Sigma-Aldrich© St. Louis, MO, USA)). The pH of the medium was adjusted with 1 N KOH to 5.7 before autoclaving at 121 °C for 20 min. Two highly-proliferative undifferentiated tamarillo ICLs, originating from two different red tamarillo genotypes, were used (L1 and L2); they were sub-cultivated in test tubes with 12 mL of jellified culture medium with the same composition as the induction medium (MS supplemented with 9% (*w/v*) sucrose and 5 mg/L of picloram) and maintained in the dark in a growth chamber at 24 ± 1 °C. ICL 1 (L1) and ICL 2 (L2) are cell lines arising from two red tamarillo genotypes.

#### 4.1.2. Establishment of Cell Suspension Cultures and Growth Kinetics 

Erlenmeyer-scaled cell suspensions were assembled to study L1 and L2 protein production and enzymatic activities. The cellular lines were grown in two liquid MS media: one with 3% (*w/v*) of sucrose and 5 mg/L of picloram and the other with 9% of sucrose (*w/v*) and 5 mg/L of picloram. Three replicates per condition were used, giving six samples for each cell line. A ratio of 500 mg of callus per 250 mL of liquid medium was used by transferring masses of proliferating callus (after four weeks of sub-culture) from the jellified medium previously described to the liquid culture media in 500 mL Erlenmeyer flasks. The cell cultures were maintained in an orbital shaker at 120 rpm in the dark at 24 ± 1 °C for two weeks.

Aliquots of 1 mL from all samples were taken in a sterile laminar flow chamber during cell growth. The first growth evaluations (optical density and number of cells) were made 1 h after growth initiation and afterward at days 7 and 14 of culture. The optical density of the suspension cultures was evaluated by measuring their absorbance at 630 nm in quartz cuvettes in a Thermoscientific Genesys 140 spectrophotometer. During growth, aliquots of cells were collected, mounted in slides, and observed using a Nikon Eclipse Ci-L optic microscope.

#### 4.1.3. Effect of Biotic Elicitors on Cell Growth

The hypothesis that the product’s yield and the final callus biomass could be enhanced was tested, subjecting the cells to the presence of biotic elicitors. Three kinds of biotic elicitors were evaluated: casein hydrolysate (CH, Sigma-Aldrich©), yeast extract (YE, Oxoid© Basingstoke, UK), and chitosan (CTS, Sigma-Aldrich©). Casein was hydrolyzed, and yeast extract was added to the culture medium in two final concentrations: 0.5 g/L and 1.5 g/L. These biotic elicitors were weighed, dissolved in distilled water, filter-sterilized, and added to the liquid culture medium after pH adjustment with 1 N KOH to 5.7 before autoclaving at 121 °C for 20 min. Chitosan was prepared according to [58] by dissolving the purified chitosan in 1% (*w/v*) acetic acid under continuous stirring. To improve the solubility of chitosan, the solution was heated at 60 °C. When dissolved, the pH of the solution was adjusted to 5.7 using 1 M NaOH 7 before autoclaving at 121 °C for 20 min. The final concentration used was 0.05 g/L and 0.1 g/L. 

Erlenmeyer-scaled cell suspensions for L1 and L2 ICL were established with a ratio of 500 mg of callus per 250 mL of MS liquid medium with three replicates for control and elicited samples. In the case of the elicited samples, the liquid culture medium was supplemented, under aseptic conditions in a laminar flow chamber, with CH, YE, and CTS at the end of the exponential phase (day 14). 

Aliquots of 1 mL from all samples were taken in a sterile laminar flow chamber during cell growth. The first growth evaluations (optical density and number of cells) were made 1 h after growth initiation and afterward at days 7, 14, and 20 of culture. The optical density of the suspension cultures was evaluated by measuring their absorbance at 630 nm in quartz cuvettes in a Thermoscientific Genesys 140 spectrophotometer. The number of cells was assessed using a Sedgewick Rafter counting chamber with 1 mL of the suspension cultures and then visualized on a Nikon Eclipse Ci-L optic microscope [59].

### 4.2. Protein Extraction

Total protein was isolated from the cultured cells at the end of the culture (day 20). The media were filtered through 0.22 μm membrane filters (Merck KGaA, Darmstadt, Germany), and the cells were weighted and preserved at −80 °C. 

For protein extraction, 200–300 mg of cells were dipped in liquid nitrogen and ground until a fine powder was obtained. Then, 5 mL of sodium phosphate buffer 0.05 M pH 7 was added, and the samples were homogenized and collected in Falcon tubes. Afterward, samples were centrifuged at 14,000 rpm at 4 °C for 15 min, and the supernatant was recovered and preserved at −20 °C.

### 4.3. Protein Quantification 

The total amount of intracellular protein was assayed using a Quick Start™ Bradford Bio-Rad^®^ Protein Assay (Bio-Rad^®^ Hercules, CA, USA) based on Bradford’s reaction [60] in a 96-well microplate as described by the manufacturer for the standard microplate assay. Briefly, 250 μL of Bradford reagent was added to 5 μL of standard or sample solutions. A calibration curve was constructed using bovine serum albumin (BSA) concentrations of 125, 250, 500, 750, 1000, 1500, and 2000 μg/mL. Absorbance measurements were made simultaneously and in triplicate at 595 nm in a SpectraMax^®^ PLUS 384 (Molecular Devices, LLC, Sunnyvale, CA, USA) spectrophotometer.

### 4.4. Enzymatic Activity Assays

The enzymatic profiles were assayed using specific activity assays based on standard procedures [61,62]. The first group of enzymatic substrates that detected the presence of proteases had the fluorogenic group amino methylcoumarine (AMC) in its C-terminal. These substrates were the amino acids methionine (L-Methionine-4-methylcoumaryl-7-amide: Met-AMC), phenylalanine (L-Phenylalanine-4-methylcoumaryl-7-amide: Phe-AMC), lysine (L-Lysine-4-methylcoumaryl-7-amide: Lys-AMC), alanine (LAlanine-4-methylcoumaryl-7-amide: Ala-AMC), leucine (L-Leucine-4-methylcoumaryl-7-amide: Met-AMC), and the complex substrate Glycyl-L-Proline-7-amide-4-methylcoumaryl (Gly-Pro-AMC) (Sigma-Aldrich^®^). Reactions took place in a 96-well microplate for fluorescence assays adding 2 μL of the substrate to 100 μL of the sample (50 μL of sodium phosphate buffer 0.05 M pH 7 and 50 μL of protein sample) using a SpectraMax^®^-Gemini™ microwell fluorescent reader (Molecular Devices, LLC) at 37 °C with an excitation and emission wavelength of 380 and 460 nm, respectively, for 40 min. In a second assay, the fluorogenic group methylumbelliferyl (MU) was used. In this case, the substrates tested were: 4-Methylumbelliferyl-β-D glucopyranoside (MU-G), 4-Methylumbelliferyl-acetyl-β-D-glucosaminide (MU-NAG), 4-Methylumbelliferyl-acetyl-β-D-phosphate (MU-P), and 4-Methylumbelliferyl-acetyl-β-Dglucosaminidase (MU-NAG) (Sigma-Aldrich^®^). MU-G, MU-NAG, and MU-C were used to detect the glycoside hydrolases whereas MU-P was used to detect the alkaline phosphatases. Once again, the reaction occurred in a SpectraMax^®^ Gemini™ microwell fluorescent reader (Molecular Devices, LLC) at 37 °C for 40 min with excitation and emission wavelength of 365 and 460 nm, respectively. The results were obtained as the slope of the reaction curve in relative fluorescent units (RFU) per second and then converted to pmol AMC or MU/min/μg of protein.

### 4.5. Statistical Analysis

The homogeneity of variances was tested with the Brown–Forsythe test (Brown and Forsythe 1974) (*p* < 0.05). In the case of homogeneity of variances, the data were analyzed with an unpaired t-test (*p* < 0.05). If three or more groups existed, one-way and two-way analysis of variance (ANOVA) was used, and where necessary, the means were compared using the Tukey test (*p* < 0.05). 

## 5. Conclusions

In summary, we showed that a 9% (*w/v*) sucrose concentration improved total protein production in the tamarillo cell suspension cultures, whereas 3%(*w/v*) sucrose enhanced specific enzymatic activity, thus 3%(*w/v*) sucrose concentration was used in the following experiments.

Moreover, at a 3% sucrose (*w/v*) concentration, tamarillo cell suspension cultures elicited with CH at 0.5 g/l and 1.5 g/l at the onset of the stationary phase tended to have a higher growth rate and protein quantity, whereas YE and CTS elicitation at the same time point showed a negative impact on these variables.

Despite that, when the specific enzymatic activity was evaluated with a fluorogenic substrate of three classes of hydrolases, the CTS-elicited cultures showed higher specific activity. 

Our findings suggest that a two-way strategy for eliciting tamarillo cell suspension cultures can be achieved using two sets of elicitors. To improve cell growth and unspecific protein production, we can use CH at both, whereas to enhance specific enzyme production, in particular, plant defense mechanisms specific enzymes, such as hydrolytic enzymes, CTS should be used. 

These findings pave the way for establishing tamarillo cell suspension cultures as a novel plant-based bioreactor system capable of transposing plant cell-based suspension cultures from a laboratory scale towards a pilot bioreactor scale with optimized production of hydrolytic enzymes with applicability in the agri-food industry.

## Figures and Tables

**Figure 1 plants-12-00190-f001:**
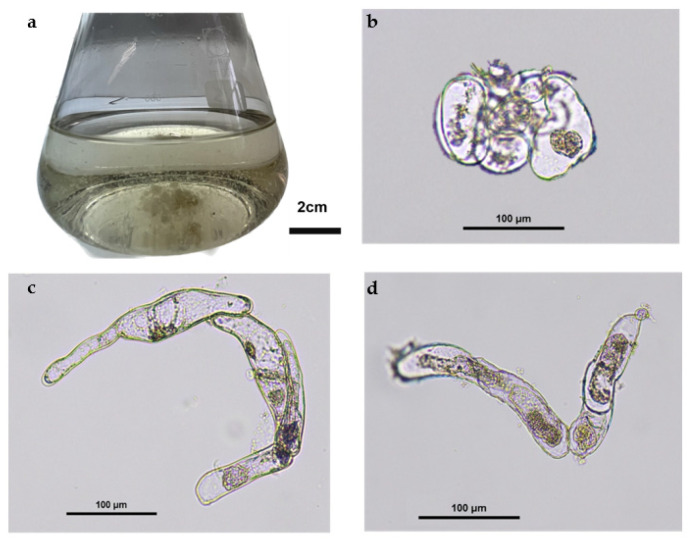
*Solanum betaceum* cell suspension cultures obtained from induced callus line 2 grown for 14 days in MS medium with 3% (*w/v*) of sucrose and 5 mg/L of picloram (**a**). Cells collected at the beginning of the cell suspension culture (day 3) (**b**) present a rounder shape, whereas cells collected at the end of the culture, on day 11 (**c**) and day 14 (**d**), present an elongated shape.

**Figure 2 plants-12-00190-f002:**
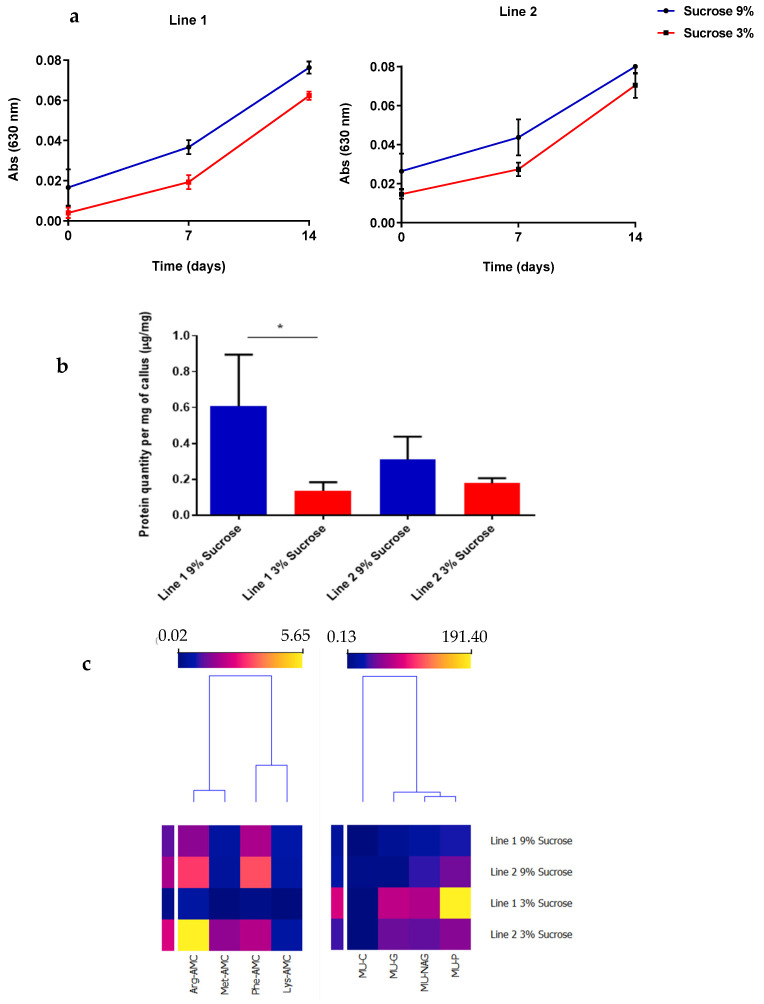
Cell growth, total protein quantity and enzymatic profiles of induced callus lines 1 and 2 with 3% and 9% (*w/v*) sucrose in culture media. (**a**) Cell growth evaluation every three days for 14 days, at different sucrose concentrations, by measuring the optical density at 630 nm. (**b**) Protein quantity per mg of callus (µg/mg), for cell lines 1 and 2, evaluated after mechanic cell lysis using liquid nitrogen and 0.05 M pH7 sodium phosphate buffer protein solubilization, and by using Bradford assay for quantification. (**c**) Clustering of enzymatic profiles of the assayed cultures using specific activity assays. The first group of specific substrates (**left**) was used to evaluate the presence of proteases. The second group of substrates (**right**) was used to evaluate the presence of glycoside hydrolases (MU-G, MU-NAG, and MU-C) and alkaline phosphatases (MU-P). Data presented as mean (*n* = 3). Statistical analysis with unpaired t test (*p* < 0.05): **** *p* < 0.0001; *** *p* < 0.001; ** *p* < 0.01; * *p* < 0.05.

**Figure 3 plants-12-00190-f003:**
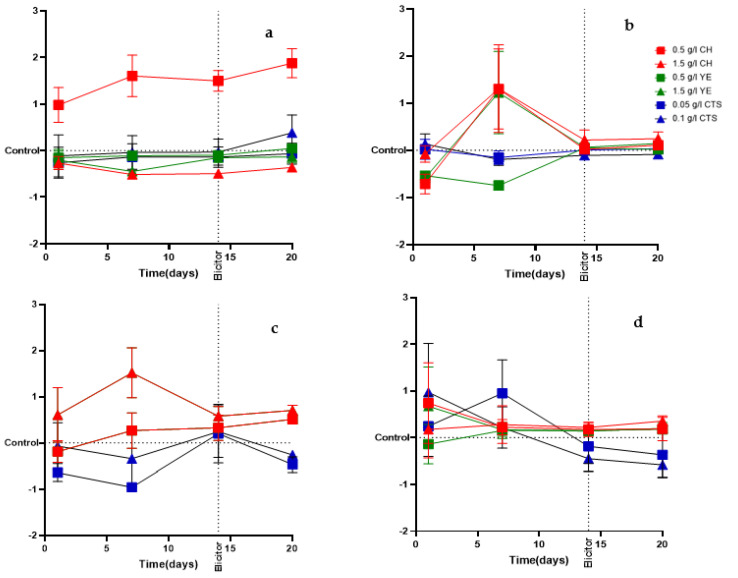
Relative cell growth to control evaluated with different elicitor treatments (CH-casein hydrolysate; YE-yeast extract; CTS-chitosan) for induced cell lines L1 and L2. Cell number measured using a Sedgwick Rafter counting chamber (L1 (**a**) and L2 (**b**)), and optical density at 630 nm (L1 (**c**) and L2 (**d**)) were measured every three days during 20 days of culture. Elicitors were applied at day 14 of culture. Data presented as mean ± SEM (*n* = 3).

**Figure 4 plants-12-00190-f004:**
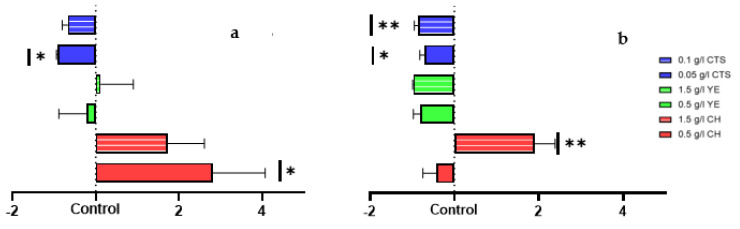
Relative total protein quantity (µg) to control, with different elicitors treatments (CH-casein hydrolysate; YE-yeast extract; CTS-chitosan), for induced callus lines 1 (**a**) and 2 (**b**), evaluated after mechanic cell lysis using liquid nitrogen and 0.05 M sodium phosphate buffer protein solubilization, and by using Bradford assay for quantification. Data presented as mean ± SEM (*n* = 3). Statistical analysis with one-way analysis of variance (ANOVA) (** *p* < 0.01; * *p* < 0.05).

**Figure 5 plants-12-00190-f005:**
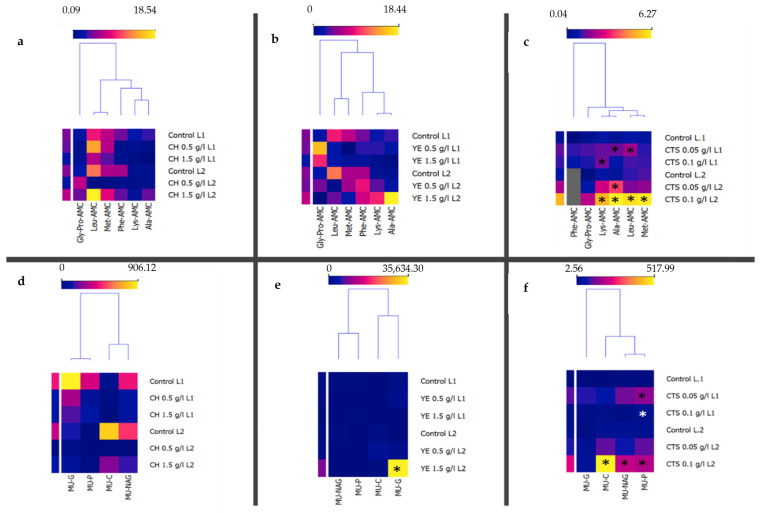
Clustering of enzymatic profiles of the assayed elicited cultures, with different treatments (CH-casein hydrolysate; YE-yeast extract; CTS-chitosan), of induced callus lines 1 (L1) and 2 (L2) using enzymatic activity assays. The first group of enzymatic substrates (**a**–**c**) was used to evaluate the presence of proteases. The second group of substrates (**d**–**f**) was used to evaluate the presence of glycoside hydrolases (MU-G, MU-NAG, and MU-C) and alkaline phosphatases (MU-P). Data presented as mean (*n* = 3). Statistical analysis with one-way analysis of variance (ANOVA) (* *p* < 0.05).

## Data Availability

The original contributions presented in the study are included in the article. Further inquiries can be directed to the corresponding author.

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
