# Peer review of "Enhancing the Production of Hydrolytic Enzymes in Elicited Tamarillo (Solanum betaceum Cav.) Cell Suspension Cultures"

_plants, 2023, doi:10.3390/plants12010190_

Round 1

Reviewer 1 Report

The present manuscript describes the production of hydrolytic enzymes in elicited tamarillo cell suspension cultures. Overall the paper is very well written and clear, but unfortunately, I wasn´t able to see the graphic results, probably due to a formatting error in the PDF version. Therefore, I suggest major modifications in the present manuscript to have the oportunity to evaluate a full version with the results.

In addition, there is a repeated sentence in line 106-110.

Author Response

We would like to acknowledge the valuable comments, corrections and suggestions raised that helped us to

improve the manuscript.

All the comments were considered, and the suggested revisions and/or corrections were made (revised word document with tracked changes).

A detailed response to the comments addressed is given below, describing the changes made to the

manuscript. The text written in blue corresponds to our answers to the comments.

Point 1: The present manuscript describes the production of hydrolytic enzymes in elicited tamarillo cell suspension cultures. Overall the paper is very well written and clear, but unfortunately, I wasn´t able to see the graphic results, probably due to a formatting error in the PDF version. Therefore, I suggest major modifications in the present manuscript to have the opportunity to evaluate a full version with the results.

Response 1: A new version of the manuscript will be submitted to avoid formatting changes

Point 2: In addition, there is a repeated sentence in line 106-110.

Response 2: The proposed corrections have been made.

Reviewer 2 Report

In this study, the effects of different sucrose concentrations on the proliferation of tomato cell suspension culture were investigated, and the activities of casein hydrolysate, yeast extracts and chilosan in promoting the activities of various proteins and enzymes were studied. Although this study was not the latest research results, this project has a good application prospect;

The overall writing of the paper is relatively standardized, there are only fewer errors, and the paper is worth publishing, but there are some places that need to be revised and can be published;

1.      Line 78 properties [16]; Line 464 [56], Pay attention to spaces;

Line 85 in vitro, in vivo should be italics;

Line 446, add a comma after [42];

2.      All the letters marked in the figures do not need a (); so it looks simple to write and the text description is much more concise;

3.      Figure 1 depicts photographs of cell suspension cells from two strains lines, the content is somewhat duplicated, and I recommend arranging several cell suspension lines cultured in the flasks here;

4.      Figures 2, 3, and 5 deviate from the normal position in the text, and the editor cannot see it clearly;

5.      Titles in the reference should be lowercase except for the first letter;

6.      Please see the attachment, I have marked the problems that appear in the paper in yellow, please note the modification.

Author Response

We would like to acknowledge the valuable comments, corrections and suggestions raised that helped us to

improve the manuscript.

All the comments were considered, and the suggested revisions and/or corrections were made (revised word document with tracked changes).

A detailed response to the comments addressed is given below, describing the changes made to the

manuscript. The text written in red corresponds to our answers to the comments.

Point 1: Line 78 properties [16]; Line 464 [56], Pay attention to spaces;

Line 85 in vitro, in vivo should be italics;

Line 446, add a comma after [42];

Response 1: The proposed corrections have been made.

Point 2: All the letters marked in the figures do not need a (); so it looks simple to write and the text description is much more concise;

Response 2: The proposed corrections have been made.

Point 3: Figure 1 depicts photographs of cell suspension cells from two strains lines, the content is somewhat duplicated, and I recommend arranging several cell suspension lines cultured in the flasks here;

Response 3: The proposed corrections have been made. Figure 1 was rearranged, by including a new photograph showing a cell suspension line cultured in a flask, and by removing some of the images previously presented to avoid duplication.

Point 4: Figures 2, 3, and 5 deviate from the normal position in the text, and the editor cannot see it clearly;

Response 4: A new version of the manuscript will be submitted in pdf format to avoid formatting changes

Point 5: Titles in the reference should be lowercase except for the first letter;

Response 5: The proposed corrections have been made.

Round 2

Reviewer 1 Report

The manuscript was adjusted as previously suggested and a new version with full results and some modifications is now presented. The manuscript still require some minor modifications suggested bellow, but overall it brings some new evidences in its field.

Line 412  On the other hand, sucrose is an essential nutrient for plant cell growth” is redundant, as this paragraph begins with this information.

Line 419 – This new paragraph should be fused with the previous, as it remains in the same subject. In addition, in line 425 the slow grow was observed in 3% sucrose COMPARED with 9% sucrose, or the slow grow was OBSERVED in the 9% sucrose treatment?

Line 512 and 515 should be fused resulting in a single paragraph from line 508.

Line 652 the fact that 9% sucrose was better for overall protein synthesis should also be noticed, and clearly state that 3% was better based on enzymatic activity.

Author Response

We would like to acknowledge the valuable comments, corrections and suggestions raised that helped us to improve the manuscript.

All the comments were considered, and the suggested revisions and/or corrections were made (revised word document with tracked changes).

A detailed response to the comments addressed is given below, describing the changes made to the manuscript. The text written in red corresponds to our answers to the comments.

Point 1: Line 412  “On the other hand, sucrose is an essential nutrient for plant cell growth” is redundant, as this paragraph begins with this information.

Response 1: The proposed corrections have been made.

Point 2: Line 419 – This new paragraph should be fused with the previous, as it remains in the same subject. In addition, in line 425 the slow grow was observed in 3% sucrose COMPARED with 9% sucrose, or the slow grow was OBSERVED in the 9% sucrose treatment?

Response 2: The proposed correction has been made and the paragraph has been clarified so that the referred slow growth is related to 3% sucrose when compared to 9% sucrose.

Point 3: Line 512 and 515 should be fused resulting in a single paragraph from line 508.

Response 3: The proposed corrections have been made.

Point 4: Line 652 the fact that 9% sucrose was better for overall protein synthesis should also be noticed, and clearly state that 3% was better based on enzymatic activity.

Response 4: The proposed corrections have been made and the paragraph has been rewritten.

Reviewer 2 Report

Since the authors have been revised all questions, I think it is accepted in current form.

Author Response

We would like to acknowledge the valuable comments, corrections, and suggestions raised that helped us to improve the manuscript. Thank you very much.